# The Evolution and Developing Importance of Fetal Magnetic Resonance Imaging in the Diagnosis of Congenital Cardiac Anomalies: A Systematic Review

**DOI:** 10.3390/jcm11237027

**Published:** 2022-11-28

**Authors:** Marios Mamalis, Ivonne Bedei, Bjoern Schoennagel, Fabian Kording, Justus G. Reitz, Aline Wolter, Johanna Schenk, Roland Axt-Fliedner

**Affiliations:** 1Division of Prenatal Medicine & Fetal Therapy, Department of Obstetrics & Gynecology, Justus-Liebig-University Giessen, 35390 Giessen, Germany; 2Diagnostic and Interventional Radiology and Nuclear Medicine, University Medical Centre Hamburg-Eppendorf, 20246 Hamburg, Germany; 3Northh Medical GmbH, 22335 Hamburg, Germany; 4Department of Cardiovascular Surgery, Justus-Liebig-University Giessen, 35390 Giessen, Germany

**Keywords:** fetal cardiac magnetic resonance, congenital heart abnormalities

## Abstract

Magnetic Resonance Imaging (MRI) is a reliable method, with a complementary role to Ultrasound (US) Echocardiography, that can be used to fully comprehend and precisely diagnose congenital cardiac malformations. Besides the anatomical study of the fetal cardiovascular system, it allows us to study the function of the fetal heart, remaining, at the same time, a safe adjunct to the classic fetal echocardiography. MRI also allows for the investigation of cardiac and placental diseases by providing information about hematocrit, oxygen saturation, and blood flow in fetal vessels. It is crucial for fetal medicine specialists and pediatric cardiologists to closely follow the advances of fetal cardiac MRI in order to provide the best possible care. In this review, we summarize the advance in techniques and their practical utility to date.

## 1. Introduction

Fetal cardiac abnormalities represent the subgroup of prenatally diagnosed abnormalities with the highest incidence (9:1000). Fetal echocardiography to date is the gold standard in fetal cardiology. Recently, reports on fetal cardiac MRI using different techniques have been published. Cardiac MRI is able to precisely demonstrate the cardiac anatomy and abnormal heart morphology in case of congenital heart disease (CHD). Furthermore, by applying the knowledge gained on fetal lambs, it can provide information about the impact of CHD on human fetal hemodynamic status by quantifying blood flow in fetal vessels [1,2,3] and provides details about fetal oxygenation [1]. In this setting, MRI has been applied to define the effect of CHD and fetal growth restriction (FGR) on brain development in fetal life [1,2] in recent studies.

In this review, the established techniques and the clinical utility of cardiac MRI in daily clinical practice will be discussed.

## 2. Fetal Cardiac MRI Techniques

Several authors refer to the utility of single-shot balanced Steady State Free Precession (bSSFP) and Fast Spin Echo sequences (SS-FSE) for detecting fetal defects [4,5,6,7,8,9,10,11]. Their contribution in providing multislice protocols is the result of a combination of high-resolution 2D static images in short time (1.0 to 1.5 mm in-plane and ≥500 ms). Although they cannot resolve fetal cardiac motion, resulting in the blurring of dynamic structures, they contribute to identify gross anatomical defects. Although bSSFP cannot preclude blurring resulting from fetal motion, it can be used for the assessment of the intracardiac anatomy as a stack of a single slice cine or 2D slices and it is capable to provide information about tissue when it is combined with SS-FE. SS-FE produces “black blood” images, useful for examining the extracardiac vasculature [11].

Furthermore, the repeated acquisition of bSSFP images in the same anatomical location over time has been used to provide a dynamic “real-time” assessment of the fetal heart including measurements of cardiac function [4,6,7,8,12,13,14]. 

Cine imaging requires the synchronization of use of high spatial and temporal resolution images to the patient’s heart rate (cardiac gating) in order to provide assessment of both cardiac anatomy and function [15]. In order to overcome the limitation of long reconstruction times, devices are developed to monitor fetal heart rate during MRI data acquisition. The most mature device for this purpose uses an MRI-compatible Doppler ultrasound gating (DUS) probe placed over the maternal abdomen [16,17]. This device monitors the blood flow and cardiac contraction based on Doppler waveform (Figure 1). The disadvantages of this method are the equipment procurement, preparation time, and monitoring of fetal lie ensuring that it is within the detection range.

The need of acceleration of acquisition led several studies to investigate the use of compressed sensing to achieve highly accelerated fetal acquisitions using either Cartesian [18,19] or radial sequences [20,21,22].

In order to limit the artifacts from maternal respiration, it has been attempted to either acquire data under maternal breath hold or apply free breathing methods using motion correction. The latter allows for multislice acquisition, in contrast to the method under maternal breath hold, which limits the number of slices [21,22,23].

With the aid of 2D, real imaging motions are recognized and data acquired during those periods are discarded. The latest fully automated methods are implied; previously used for the fetal brain volumetric reconstruction. The outlier rejection, results in better quality images by estimating the probability of each voxel and real-time image frame, classifying them as in- or outlier, and, finally, rejecting the outlier voxels and frames from the final CINE reconstruction [23,24,25].

To achieve a 3D demonstration of the fetal heart and extracardiac vasculature [26], and a 4D whole-heart visualization [23], the combination of 2D images and multi-planar acquisition using volumetric reconstruction methods has been applied. For instance, a 4D flow cine MRI reconstruction was achieved by exploiting the velocity-sensitive information inherent to the phase of dynamic bSSFP acquisitions in combination with signal-to-voice ratio (SVR). Multiple non-coplanar bSSFP stacks were used to reconstruct spatially identical, temporally resolved, motion-corrected magnitudes and vector flow volumes [24]. The advantages of this method represent the robustness in motion, the capability of both in-plane and through-plane motion to be corrected, and the full coverage of the fetal heart.

In 2010, an artificial cardiac triggering, or self-gating system, was proposed to the so-called Metric Optimized Gating (MOG). It detects mis gating artifact through evaluation of image metrics while images are retrospectively analyzed. Whereas this method was first developed for time-resolved phase-contrast measurements of fetal blood flow, it has since been applied to both Cartesian and radial bSSFP acquisitions and was the first method to demonstrate dynamic CINE MRI of the human fetal cardiovascular system [27,28]. MOG, however, does not address fetal or maternal body motion.

To achieve multidimensional flow in fetuses, a golden-angle radial phase-contrast cardiovascular magnetic resonance has been applied with real-time reconstructions; first performed for retrospective motion correction and cardiac gating using MOG [29].

Another approach to fetal cardiac gating is MRI self-gating, where a periodic gating signal is extracted from the MRI data itself and it is used to sort the data retrospectively [30].

## 3. Current Potential and Clinical Application of Fetal Cardiac MRI

A cardiovascular fetal MRI was first attempted in 2005 when applying real-time sequences with the aim of estimating the ventricular volume. Small case series of investigating the normal heart anatomy and congenital heart defects (CHD) followed and although some reviews have attempted to establish protocols [31], there is no consensus on which are the indications of a cardiovascular MRI. Manganaro et al. performed fetal MRI in 32 fetuses with a mean gestational age of 30 weeks from January 2007 to March 2008 with a US-assessed CHD. All the morpho-volumetric abnormalities of the heart, cardiac axis abnormal rotation, ventricular septal defects (VSDs) as well as cases with abnormal origin and course of the great vessels (GV) could be confirmed by MRI using SSFP sequences. In Figure 2, a hypoplastic right heart and a VSD in 4CV are illustrated utilizing cine SSFP in a late-term fetus in the overdiagnosis of two VSDs, the restricted evaluation of atrial septum (AS) due to the presence of foramen ovale (FO) and its low thickness, the restricted evaluation of the size of GV due to MRI spatial resolution and SNR, and the insufficient evaluation of valves have proven that although MRI is a useful adjunct, improvements in gating, speed of sequences, and signal-to-noise improvement were necessary at that time.

Over next years, some of the improvements have been achieved and cardiac MRI is able to detect the patency of the interatrial septum that is crucial information for the delivery and treatment plan for Transposition of Great Arteries (TGA) fetuses [32,33]. In a large study of Su-Zhen Dong et al., MRI was applied as an adjunct to classic fetal US, when the latter was not able, due to common known difficulties, to make a conclusive diagnosis [34]. Over 14 years, 71 cases with confirmed CHD have been reviewed applying a multi-planar reformatting reconstruction 2D SSFP technique to reconstruct the fetal cardiovascular MR images. In 60.6% of cases, MRI put a conclusive correct diagnosis and only 4.2% of cases were correct but incomplete. However, the author considers some types of defect, such as unclosed Ductus Arteriosus (DA), at birth or secundum Septal Defect (SD) as impossible to diagnose prenatally. For complex CHD, such as Pulmonary Atresia with intact ventricular septum (PA/IVS), severe Tetralogy of Fallot (TOF), Double Outlet Right Ventricle (DORV), or Transposition of Great Arteries (TGA) a definitive diagnosis could not be made by use of MRI. Finally, mild Pulmonary Stenosis (PS) and small Ventricular Septal Defect (VSD) are missed by cardiac MRI.

CMR is a useful adjunct for the evaluation of the cardiac vascular anatomy and several attempts have been made to study the aortic arch, its branches, and its anatomical position in relation to the trachea; in particular when vascular rings are suspected. Similarly, as regards the blood supply of the lungs challenging for echocardiography cases, such as pulmonary atresia with MAPCAs or pulmonary trunk can be elucidated with the aid of MRI. The drainage of both pairs of pulmonary veins is feasible to be demonstrated by MRI as well.

Su-Zhen Dong et al., in their in 2020 review, using data from 71 cases with confirmed CHD over 14 years, concluded that the Coarctation of Aorta (CoA) prenatal diagnosis was unreliable due to both the technical difficulty of imaging this region and the obligatory patency of the arterial duct in fetal life and physiologic changes of the DA after birth [34]. Su-Zhen Dong concluded in a previous study in 2018, after having examined six fetuses of mean gestational age (26.5 weeks) with Aortic Arch (AoA) abnormality using bSSFP and SSFE techniques, that fetal cardiac MRI is a useful complementary tool to assess fetuses with right aortic arch and right ductus arteriosus. In half of the cases, the aortic arch laterality abnormality was missed in prenatal US [35]. In the same line, the same author reported that MRI was able to demonstrate the abnormal course of left brachiocephalic vein in nine fetuses in a retrospective review of 7282 fetuses from June 2006 to March 2017 [36] and a LPSVC in 49 fetuses in a retrospective review from January 2010 to October 2015 [37]. A recent study of Ryd et al. has proven, after having investigated 31 fetuses at a median age of 31 weeks, that CMR had clinical utility affecting patient management and/or parental counselling in 26 cases (84%) [38]. For assessment of univentricular vs. biventricular outcome in borderline left ventricle, unbalanced atrioventricular septal defect and pulmonary atresia with intact ventricular septum (15 fetuses) fetal CMR visualized intracardiac anatomy and ventricular function, allowing assessment of outcome in 13 cases (87%). In four fetuses with hypoplastic left heart syndrome, it helped delivery planning in three of those cases (75%). For aortic arch anatomy including signs of coarctation (20 fetuses), CMR added diagnostic information in 16 cases (80%), which is not in line with the previously mentioned study of Su-Zhen Dong et al. [34]

Lοyd et al., by applying novel MRI with high-resolution motion-corrected three-dimensional volumes of the fetal heart and phase-contrast flow sequences gated with metric-optimized gating on 51 fetuses with suspected CoA, concluded that MRI with the aid of a multivariate logistic regression model, including aortic flow and isthmic displacement, may have an important role in predicting severe neonatal CoA outcome and need for intervention in 93% of cases [39]. In the same line is another prospective single-center cohort study of Lyod et al. in which 85 fetuses with suspected CHD have been examined from October 2015 to June 2017 using MRI with motion-corrected slice-volume registration. The data as overlapping stacks of 2D images were processed with a bespoke open-source reconstruction algorithm to produce a super-resolution 3D volume of the fetal thorax. Vascular measurements, although showed good overall agreement with 2D echocardiography in 51 cases, fetal vascular structures have been more effectively visualized with 3D MRI and have had a higher diagnostic quality score [27]. Finally, Xu Li et al. compared the accuracy in correct diagnoses of aortic arch anomalies of both fetal US and CMR using SSFP and SSFSE in 600 pregnant women from January 2013 to 22 December 2015 of them with aortic arch anomalies. MRI revealed a 95.6% accuracy misdiagnosing only one case of right aortic arch with aberrant left subclavian artery as a double aortic arch. The same case has been misdiagnosed by US, which showed an accuracy of 60.8% [40].

Apart from cardiovascular features, the application of cardiac MRI can play an additive role on extracardiac anatomy, such as laterality disorders and lung parenchyma characterization in cases of pulmonary venous obstruction or intact interatrial septum in which lymphangiectasia is suspected [41,42]. Elisabeth Mlczoch et al. investigated fetal lung volume (TLV) in 105 fetuses with CHD and mean gestational age 26 + 6 weeks of gestation from January 2004 to December 2011 by applying fetal cardiac MRI SSFP and FSE only axial T2 sequences, concluding that fetuses with CHD had significantly smaller TLV in comparison with non-CHD fetuses and that small pulmonary arteries correlate with small TLV. A total of 17% of fetuses had TLV below normal indicating pulmonary hypoplasia [43].

Saul et al. investigated 44 fetuses with HLHS to identify which ones developed Lymphangiectasia described as nutmeg lung in MRI; the nutmeg lung MR appearance in HLHS fetuses is associated with increased mortality/OHT (100% in the first 5 months of life compared to 35% with HLHS alone). Not all patients with restrictive lesions develop nutmeg lung, and the outcome is not as poor when restriction is present in isolation. Dedicated evaluation for nutmeg lung pattern on fetal MR studies may be useful in guiding prognostication and aiding clinicians in counselling parents of fetuses with HLHS [44] 

Investigations into the circulatory physiology of fetal sheep were undertaken in the 1930s by Sir Joseph Barcroft at Cambridge University. Sir Geoffrey Dawes conducted the first detailed studies of the fetal circulation. ΜRI spectroscopy at high field strengths and BOLD (Blood Oxygenation Level Dependent) signal allowed for information about blood and tissue oxygenation by relating the T2 and T2* transverse-relaxation time of blood and the oxygenation state of hemoglobin on erythrocytes. Although BOLD does not provide quantitative data about blood oxygen saturation, it is possible to estimate both hematocrit and oxygen saturation by combining T1 and T2 measurements of blood. Sun et al., by applying MOG for fetal triggering and using an accelerated version of the acquisition utilized by Wedergartner’s fetal lambs, which incorporates a motion correction algorithm, were able to make reproducible measurements of T2 in the larger fetal vessels in 40 late-gestation human fetuses [45].

Moreover, the relationship of fetal hemodynamics to brain and lung development in cases of fetal growth restriction (FGR) and complex CHD could be demonstrated by several studies [2]. Successively, MRI plays a role in studying hemodynamics. In Figure 3, 2D phase-contrast MR angiography is illustrated using Doppler ultrasound gating at 1.5 Tesla in a late term fetus (gestational age: 30 + 4 weeks) for the assessment of blood flow hemodynamics. In the intrauterine peri-operational period in cases of CHD, such as HLHS, it can investigate the grade of pulmonary obstruction. Furthermore, it detects the patency of foramen ovale (FO) indirectly by estimating the higher oxygen saturation in the Left Ventricle (LV) resulting from the preferential streaming of oxygenated blood through FO.

From 2010 to 2012, Bahiyah al Nafisi et al. investigated the fetal circulation distribution in 22 fetuses with left-sided CHD at a mean of 35 weeks of gestation using phase contrast MRI and compared them with twelve normal fetuses. Fetuses with left-sided CHD had a mean combined ventricular output (CVO) that was 19% lower than normal controls. In fetuses with left-sided CHD with pulmonary venous obstruction, pulmonary blood flow was significantly lower than in those with left-sided CHD without pulmonary venous obstruction. All 3 fetuses with pulmonary venous obstruction had pulmonary lymphangiectasia. Fetuses with small but apex-forming left ventricles with left ventricular outflow tract or aortic arch obstruction had reduced ascending aortic (AAo) and FO flow compared with normal [46]. The reference ranges of the blood flow in the GV of a fetus in the late gestation have been established with the aid of MRI [47]. Recently, Roberts et al. presented a novel method for in utero whole-heart fetal 4D cine quantitative blood flow imaging. A 4D flow cine MRI reconstruction was possible by utilizing the velocity-sensitive information inherent to the phase of dynamic bSSFP and SVR. Multiple non-coplanar bSSFP stacks were used to reconstruct spatially identical, temporally resolved, motion-corrected magnitude, and vector flow volumes [24].

## 4. Conclusions

Fetal cardiac MRI has been studied, has evolved and has an adjunctive role to fetal US, which has remained the gold standard over the last two decades. Its place, where the US cannot overcome its known limitations, has recently become of interest to several researchers. Its evolution will establish it as a necessary tool for the correct and complete diagnosis and comprehension of the anatomy and pathophysiology of CHD, the proper prenatal counselling for the outcome, and the postnatal or even prenatal operative preparation of individuals.

## Figures and Tables

**Figure 1 jcm-11-07027-f001:**
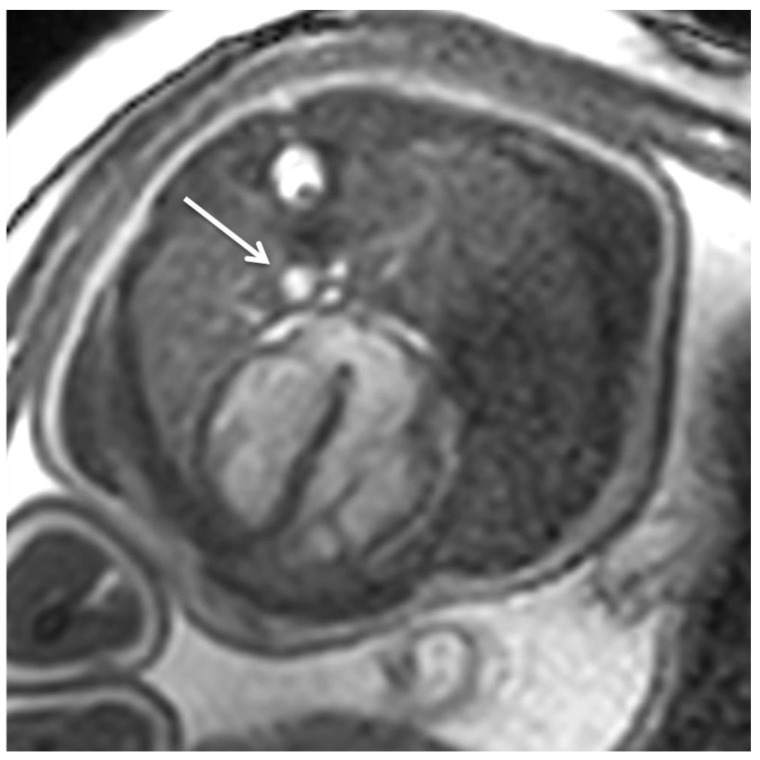
Fetal cardiac MRI using Doppler ultrasound gating at 3 Tesla: The applied rapid gradient-recalled echo (GRE) steady-state free-precession (SSFP) sequence (TR = 3.9 ms, TE = 1.9 ms, field of view = 246 × 246 mm, flip angle = 60°, slice thickness = 6 mm, matrix size 164 × 164) is adopted from standard adult cardiac MRI protocols. SSFP sequences have short acquisition times and provide high myocardium-blood contrast (composed of both T1 and T2 contributions). Fetal cardiac gating using a Doppler ultrasound sensor enables dynamic/cine imaging by acquisition of multiple images (i.e., heart phases) over the cardiac cycle. The example illustrates an end-diastolic four-chamber SSFP image and reveals physiological situs, cardiac chambers, and left descending aorta (arrow) in a healthy late term fetus (gestational age: 35 + 0 weeks). Morphometry: LCD = 441 mm, TCD = 387 mm, LVmidtransverse = 152 mm, LVlongitudinal = 261 mm, RVmidtransverse = 158 mm, RVlongitudinal = 247 mm, mitral valve plane = 86 mm, tricuspidal valve plane = 91 mm.

**Figure 2 jcm-11-07027-f002:**
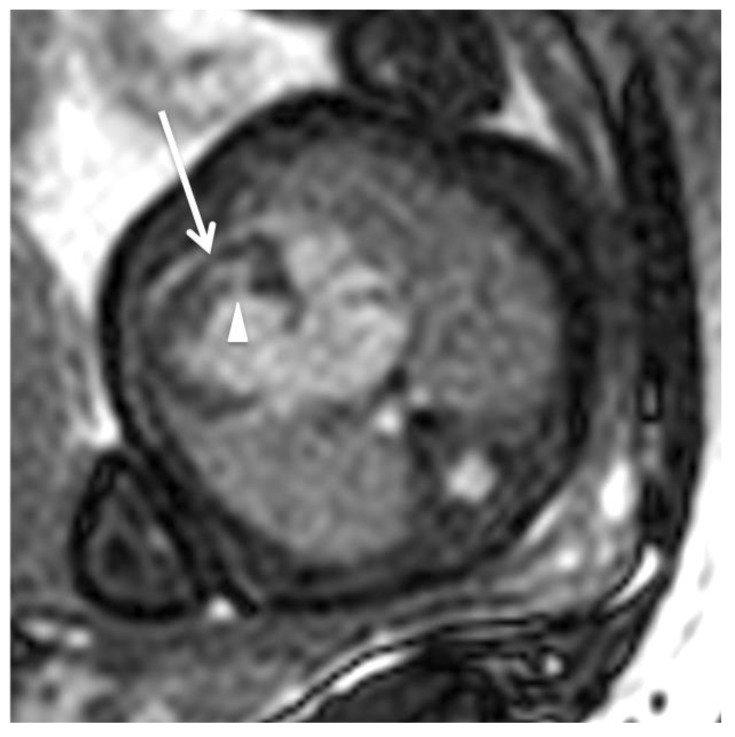
Fetal cardiac MRI using Doppler ultrasound gating for assessment of dynamic/cine steady-state free-precession (SSFP) sequences (TR = 3.3 ms, TE = 1.6 ms, field of view = 300 × 300 mm, flip angle = 60°, slice thickness = 5 mm, matrix size 288 × 288) at 1.5 Tesla: The end-diastolic four-chamber SSFP image illustrates hypoplastic right heart (arrow) and ventricular septal defect (arrowhead) in a late term fetus (gestational age: 30 + 5 weeks). Postnatal diagnosis revealed atresia of the tricuspid valve, malposition of the great arteries, and interrupted aortic arch (not visualized in the presented four-chamber view).

**Figure 3 jcm-11-07027-f003:**
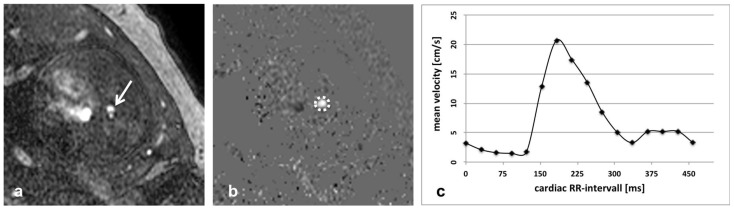
2-D phase-contrast MR angiography (TR = 4.9 ms, TE = 3 ms, field of view = 250 mm, flip angle = 15°, slice thickness 5 mm, matrix size = 194 × 194 mm) using Doppler ultrasound gating at 1.5 Tesla for assessment of blood flow hemodynamics. Transversal orientation through the descending aorta (arrow in **a**) with resulting magnitude image (**a**) and phase image (**b**). Blood flow hemodynamics are assessed by placement of a region of interest (ROI; dotted circle in **b**) in the vessel lumen. Dynamic blood flow over the whole cardiac cycle is encoded by signal intensities within the ROI. (**c**) In this example of a late term fetus (gestational age: 30 + 4 weeks) resulting mean flow velocity illustrates typical arterial flow waveform over the cardiac cycle.

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
