# Peer review of "The Evolution and Developing Importance of Fetal Magnetic Resonance Imaging in the Diagnosis of Congenital Cardiac Anomalies: A Systematic Review"

_jcm, 2022, doi:10.3390/jcm11237027_

Round 1
Reviewer 1 Report
The authors detail the usefulness of intrapartum fetal MRI scans for congenital heart disease. The fetal heart has been examined by ultrasound for some time. However, this paper discusses the new field of fetal cardiac MRI.
As mentioned in the introduction, fetal MRI has the drawback that fetal motion is difficult to control. This review discusses the latest MRI techniques for such problems. In particular, cardiac MRI, compared to fetal brain MRI, for example, has a poor image resolution due to the added motion of the heart. This point is also discussed and will be of further interest to readers.
 Unfortunately, however, none of the fetal MRI images are included in this review. Reviewer hope that it is better to include MRI images in the paper, even in a systematic review. This does not meet the reader's expectations. What kind of fetal cardiac MRI images are shown in the text, e.g., fetal cardiac MRI images of hypoplastic difference syndrome? Can you present fetal MRI images of pulmonary artery obstruction with suspected congenital lymph-angiectasia? Also, are there fetal MRI images of the more frequent atrial septal defects? Are fetal MRI images of complex cardiac malformations available?
 It would be more useful to the reader if Figure images could be presented in comparison to the ultrasound presentation to show that these images would be more useful.
 In summary, we request that the reviewer present a substantial number of images for a comprehensive review of diagnostic imaging, but we consider the absence of any fetal MRI images in this review to be a major shortcoming.
Best regards,
Dr. Reviewer
Author Response
Thank you for your recommendation. We have added MRI images that they surely meet the readers' expectations.
Reviewer 2 Report
The purpose of the study is to review the use of cardiac MRI in the evaluation of congenital cardiac anomalies.
I would suggest:
- to divide the work in more paragraphs to better understand the work;
- to do an extensive editing of english language;
- to review the number of references because are not in correct order (why in the introduction you start with 28-29?)
- to create a table with the articles cited in the article;
- to add some images.
Author Response
Thank you for your recommendations. An editing of english has been done and the article has been divided in more paragraphs to become more comprehensive. The numbers of references are now in the correct order and images have been included to meet the expectations of the readers. The citated articles are the articles in references so that additional table has not been created.
Yours sincerely
Marios Mamalis
Round 2
Reviewer 1 Report
Reviewers have reviewed the newly revised paper. The authors have added new Figures 1, 2, and 3. We believe that the revised paper will provide JCM readers with a better understanding of fetal MRI. We add our comments below and invite you to review the revisions again. The reviewers look forward to seeing the revised manuscript resubmitted.
Minor1
Please provide a radiological description of the MRI images in Figures 1 and 2. What exactly is the method of imaging? Is this spin echo imaging?
Please provide an additional radiological sequence for each MRI image. For example, is this a T2-weighted or T1-weighted image? And other than that?
Finally, for each MRI photo, what are the settings for echo time (TE) and repetition time (TR)? I will provide a reference paper for your description.
"Callosal agenesis followed postnatally after prenatal diagnosis. Congenital Anomalies 2006; 46, 160-162. doi:10.1111/j.1741-4520.2006.00120.x
Minor2
This paper also describes the importance of pre-test counseling for MRI examinations for pregnant women. This is because a fetal MRI scan may incidentally reveal a major disease. Please use this paper as a reference for additional description of informed consent in medical testing ethics.
Best regards,
Dr. Reviewer
Author Response
Dear Dr.,
Thank you for the valuable comments on the manuscript. Regarding the first recommendation all requested minor changes have been included in the manuscript.
Thank you very much for the friendly comment (minor change2)
Yours sincerely
Reviewer 2 Report
The manuscript is sufficiently improved.
Author Response
Thank you for your valuable comments.
Yours sincerely